# Phosphorus-activated carboxyl small molecule positive electrode for high specific capacity and long-life iron-organic batteries

Yehui Zhang[1], Qi Huang[2], Pingxuan Liu[1], Yaokang Lv[3], Ziyang Song [1,4], Lihua Gan [1,5] & Mingxian Liu [1,5] ✉

Iron-ion batteries represent a compelling energy storage solution due to the cost-effectiveness, suitable redox potential, and high capacity of Fe negative electrodes. Polyaniline positive electrodes for iron-ion batteries have demonstrated promising electrochemical redox properties, but face limited redox-accessible groups and unstable −NH− sites. Here we show phosphorus redox activity in a carboxyl small molecule electrode. 4,4′,4″-phosphanetriyl-tribenzoic acid and 4,4′,4″-nitrilotribenzoic acid are designed via modulating the electron-donating P and tert-N motifs, showing tuned charge distributions and energy levels. With the decrease of the electronegativity and energy barrier (N > P), 4,4′,4″-phosphanetriyltribenzoic acid exhibits stronger $Fe^{2+}$ coordination with carboxyl sites, and brings closed $CF_3SO_3^-$ proximity to P centers. This feature ensures high activity of carboxyl/phosphorus sites with low activation energy (0.24 $vs.$ 0.29 eV for 4,4′,4″-nitrilotribenzoic acid). 4,4′,4″-phosphanetriyltribenzoic acid with P-extended conjugated structure achieves low energy gap (2.28 eV) compared to its individual carboxyl or P-containing counterparts (2.71/3.16 eV), thereby enabling high utilization of carboxyl/P motifs (98.5%) and enhanced redox voltage (0.8 V). A stable 4 e⁻ $Fe^{2+}/CF_3SO_3^-$ storage of 4,4′,4″-phosphanetriyltribenzoic acid positive electrode endows Fe battery with high specific capacity (276 mAh g⁻¹) and cycling stability (60,000 cycles). This work highlights the potential of phosphorus-active organic materials toward iron-ion batteries.

The rapid development of sustainable energy storage technologies has accelerated the research on next-generation rechargeable aqueous batteries, which offer inherent safety, environmental sustainability and cost-effectiveness[1–6]. Among various aqueous batteries employing multivalent transition metal negative electrodes (such as Zn, Mg, Al, Fe)[7–11], zinc-ion batteries are prevailing systems owing to their high theoretical specific capacity (820 mAh g⁻¹/5855 mAh cm⁻³), low redox potential (−0.76 V $vs.$ standard hydrogen electrode) and relatively low cost (~2.5 USD per kg) of Zn negative electrodes[12,13]. In contrast, iron-ion batteries (IIBs) represent a compelling avenue for energy storage solutions, benefiting from the even higher theoretical specific capacity (960 mAh g⁻¹/7557 mAh cm⁻³), moderate $Fe^{2+}/Fe$ redox potential

[1]Shanghai Key Lab of Chemical Assessment and Sustainability, School of Chemical Science and Engineering, Tongji University, 1239 Siping Rd., Shanghai, P. R. China. [2]Institute for Electric Light Sources, School of Information Science and Technology, Fudan University, 2005 Songhu Rd., Shanghai, P. R. China. [3]College of Chemical Engineering, Zhejiang University of Technology, 18 Chaowang Rd., Hangzhou, P. R. China. [4]State Key Laboratory of Pollution Control and Resource Reuse, College of Environmental Science and Engineering, Advanced Research Institute, Tongji University, 1239 Siping Rd., Shanghai, P. R. China. [5]State Key Laboratory of Cardiovascular Diseases and Medical Innovation Center, Shanghai East Hospital, School of Medicine, Tongji University, 150 Jimo Rd., Shanghai, P. R. China. ✉e-mail: liumx@tongji.edu.cn

($-0.44$ V $vs$. SHE) and lower cost (about 40 times cheaper than Zn) of Fe negative electrodes[14,15]. Thus, the rational design of well-matched positive electrode materials is a crucial task to fully unleash the potential of IIBs. Currently, inorganic positive electrode materials have been reported for IIBs, such as $V_2O_5$, $VOPO_4 \cdot 2H_2O$, $LiFePO_4$ and Prussian blue analogs[16–18], which generally face sluggish kinetics and poor cyclic stability caused by the strong electrostatic interactions between $Fe^{2+}$ ions and hosting lattices.

Beyond inorganic compounds, π-aromatic organic materials exhibit great potential as positive electrodes for propelling iron-organic batteries (IOBs) owing to their flexible molecular configurations and effective structural/functional tunability[15,19]. Consequently, designing suitable and robust organic positive electrodes to efficiently store $Fe^{2+}$ ions is essential for IOBs. To this end, there have been a few preliminary investigations on organic positive electrode materials for IOBs. For example, the Li group reported a cross-linked polyaniline positive electrode with enhanced electronic conductivity and stability for activating IOBs, affording high specific capacity (209 mAh g$^{-1}$) and cycling durability (39,000 cycles)[15]. Furthermore, the Yu group developed a sandwich-type polyaniline positive electrode to demonstrate high specific capacity (225 mAh g$^{-1}$) and long lifespan (27,000 cycles)[19]. Despite these promising achievements made in broadening the battery horizons, the twisted polymeric chains and unstable structural linkages often result in disordered stacking structures, limiting the full utilization of redox-active motifs and the cycling durability of polymer positive electrodes[20–24]. Therefore, more attempts are still needed to unlock the electrochemical limitations of IOBs by strategic structural engineering.

As is well-established, designing multiple redox-active sites in organic small molecule positive electrodes that permit more electron transfer is key to boosting the capacity of IOBs[25–29]. Meanwhile, tuning the π-conjugated electronic structures of organic skeletons can regulate electron delocalization and structural robustness by lowering energy levels to afford stable charge storage[30–33]. Currently, polyanilines as prevailing positive electrodes have exhibited promising redox electrochemical performances, but are hindered by the limited redox accessibility and inherent instability of p-type −NH− sites due to twisted structure and high energy barrier, which restrict both capacity and cycling durability[15,19]. These limitations motivate us to explore better small molecules with high activity and stability of p-type sites for IIBs through rational structure engineering. Featured with strong electroactivity, desirable redox kinetics, and anion-coupling sensitivity of π−conjugated electron-donating C−P sites in aromatic rings, p-type phosphorus compounds represent promising stable positive electrode candidates, yet remain underexplored in battery fields. However, the intrinsically low density of redox-active phosphorus units in organics inevitably limits the capacity of IOBs. Grafting multi-electron-withdrawing n-type carboxyl motifs (−COOH) into phosphorus compounds can effectively tune the electronic structure and delocalization behaviors, thereby boosting electrochemical activity and durability[34–36]. Considering the complementary advantages of multi-active carboxyl and stable phosphorus motifs, designing phosphorus-activated carboxyl molecules with easy-accessible redox sites and stable structures will exhibit a synergistic vitality to achieve high specific capacity and durable IOBs, but have not yet been reported.

In this work, we unlock phosphorus redox chemistry in a small molecule carboxyl positive electrode for IOBs. 4,4′,4″-Phosphanetriyltribenzoic acid (PTBA) and 4,4′,4″-nitrilotribenzoic acid (NTBA) organic molecules are designed by modulating the electron-donating P and tert-N groups, which show tuned charge distributions and energy levels. The lower electronegativity and energy barrier (N > P) in PTBA strengthen low-potential $Fe^{2+}$-carboxyl coordination and activate high-potential $CF_3SO_3^-$-phosphorus reaction. This feature ensures high redox activity of carboxyl/phosphorus sites with a low activation energy (0.24 $vs$. 0.29 eV for NTBA). PTBA with P-extended conjugated structure shows the lowest energy gap (2.28 eV) compared to its individual carboxyl or P-containing counterparts (2.71/3.16 eV), thus giving high utilization of carboxyl/P motifs (98.5%) and enhanced redox voltage (0.8 V). Besides, the phosphorus-linked π-extended aromatic framework effectively suppresses the dissolution of the PTBA positive electrode in aqueous electrolytes to achieve stable 4 e$^-$ $Fe^{2+}$/ $CF_3SO_3^-$ storage. Fe||PTBA battery harvests the enhanced performance in terms of high specific capacity, high-rate capability and durable lifespan. This study opens good avenues for designing phosphorus-functionalized organic materials with favorable redox activity and stability for next-generation iron batteries.

## Results and discussion
### Materials characterization
Figure 1a illustrates the π-conjugated bipolar molecule skeletons of PTBA and NTBA, which share similar aromatic triple-carboxyl structures but different p-type centers (C−P $vs$. C−N). For comparison, 1,3,5-tris(4-carboxyphenyl)benzene (TCB) with electron-accepting n-type carboxyl groups and triphenylphosphorus (TPP) with electron-donating p-type phosphorus species were selected as the model molecules to evaluate their electrochemical behaviors (Supplementary Fig. 1). The morphological and structural properties of the four organic molecules were analysed by Fourier transform infrared spectra (FT-IR, Supplementary Fig. 2), scanning electron microscopy images (SEM, Supplementary Fig. 3), elemental maps (Supplementary Fig. 4), and X-ray diffraction patterns (XRD, Supplementary Fig. 5). Additionally, optimized molecular geometries[37,38], were achieved through computational simulations to further investigate their electronic configurations and redox properties (Supplementary Fig. 6, Supplementary Data 1).

By modeling the charge distribution of molecular electrostatic potentials (MEP)[39–41], the redox-bipolar-active centers of PTBA and NTBA can be identified. MEP distribution shows that the three carboxyl groups with electronegative potentials (red regions) are n-type motifs capable of chelating cations. Conversely, C−P or C−N centers with electropositive potentials (blue regions) serve as p-type sites for anion coordination (Fig. 1b, Supplementary Fig. 7). Notably, comparative MEP analysis indicates that the C−P center in PTBA possesses more negative MEP values compared to C−N in NTBA, suggesting enhanced redox activity and good ion-coupling capability. The energy gap ($\Delta E$) between the highest occupied molecular orbital (HOMO) and the lowest unoccupied molecular orbital (LUMO) is a crucial indicator for evaluating molecular electronic properties[42–44]. Of note, PTBA exhibits a narrow $\Delta E$ of 3.75 eV, which is lower than that of NTBA (4.16 eV, Fig. 1c) and unipolar TCB/TPP molecules (4.66/4.63 eV, Supplementary Fig. 8). Such a result indicates the good structural stability and charge-transfer efficiency of PTBA.

FT-IR spectra exhibit the presence of C−P groups at 745 cm$^{-1}$, and the asymmetric/symmetric stretching vibrations of C=O/−COO at 1685/1410 cm$^{-1}$ in PTBA, indicating its redox bipolarity (Fig. 1d). Moreover, the optical energy gaps ($E_g$) were calculated to be 2.28 and 2.52 eV for PTBA and NTBA (Fig. 1e, 2.71/3.16 eV for TCB/TPP, Supplementary Fig. 9), signifying their high electron conductivity to propel high-kinetics redox reactions[45,46]. Compared with NTBA ($7.80 \times 10^{-9}$ S cm$^{-1}$), PTBA with the extended π-conjugated aromatic structure shows an enhanced electrical conductivity ($8.49 \times 10^{-9}$ S cm$^{-1}$, Supplementary Fig. 10). It is further boosted to 27.05 S cm$^{-1}$ with the addition of 30 wt% acetylene black conductive agent (Fig. 1f, Supplementary Fig. 11), facilitating efficient electron transfer. Furthermore, UV-Vis spectroscopy measurements reveal no absorption signals for PTBA after immersion in Fe(OTF)$_2$/H$_2$O electrolyte (OTF$^-$ = CF$_3$SO$_3^-$), demonstrating strong resistance to dissolution (Supplementary Fig. 12). Overall, PTBA integrates rich carboxyl/ phosphorus motifs, good coordination capability, low energy barriers, and robust π-conjugated frameworks, making it an ideal positive electrode candidate for Fe batteries.

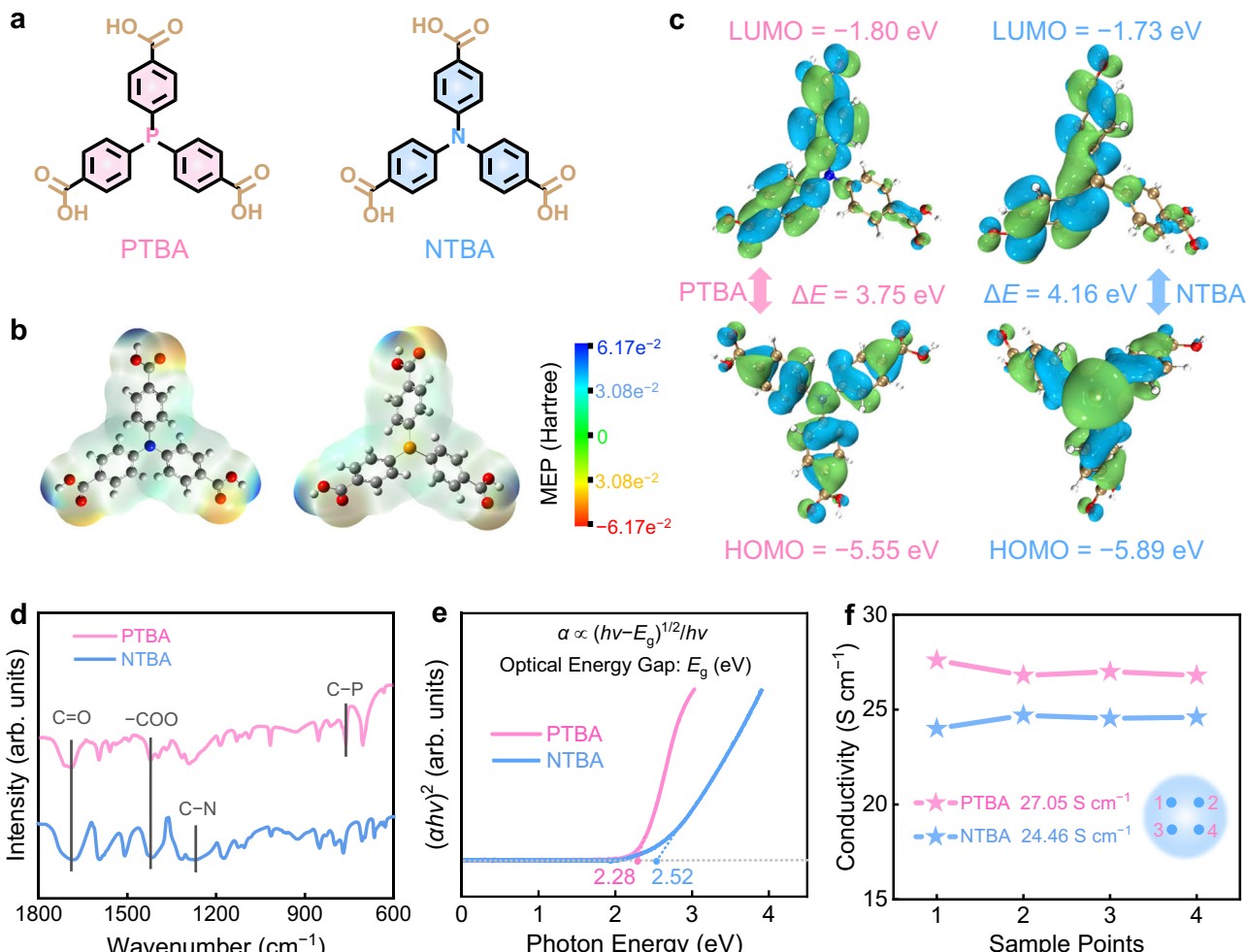

**Fig. 1 | Material characterizations. a** Bipolar molecular structures. **b** Molecular electrostatic potentials maps. Colors of elements: C, gray; O, red; H, white; P, blue; N, yellow. **c** LUMO/HOMO energy levels. Blue regions: positive phase of the orbital wave function; Green regions: negative phase of the orbital wave function. **d** FT-IR spectra. **e** Calculated energy gaps. **f** Electronic conductivities were tested at 4 different points on PTBA and NTBN positive electrodes with an RTS-8 four-point probe.

## Electrochemical performance

The Fe||PTBA cell was further assembled by using PTBA positive electrode paired with Fe metal negative electrode in aqueous 1 mol L$^{-1}$ Fe(OTF)$_2$ electrolyte (Supplementary Figs. 13–15). Three identical Fe||PTBA cells were tested for a single electrochemical experiment. Acetylene black as conductive additive demonstrates a negligible specific capacity (8 mAh g$^{-1}$) for Fe||PTBA battery (Supplementary Fig. 16). Two well-defined redox peaks in cyclic voltammetry (CV) profiles signify a two-step redox reaction of PTBA positive electrode (Fig. 2a). Notably, owing to the favorable LUMO-HOMO gap and high electronic conductivity, PTBA positive electrode demonstrates enhanced redox potentials compared to NTBA positive electrode, resulting in an elevated average discharge voltage of 0.8 V. Galvanostatic charge-discharge (GCD) profiles of Fe||PTBA battery exhibit obvious two discharge voltage plateaus consistent with the two-step redox behavior (Fig. 2b, c), providing good reversible capacities (276 mAh g$^{-1}$@0.5 A g$^{-1}$ and 141 mAh g$^{-1}$@20 A g$^{-1}$), which exceed that of Fe||NTBA battery (237 mAh g$^{-1}$@0.5 A g$^{-1}$ and 106 mAh g$^{-1}$@20 A g$^{-1}$).

Furthermore, the almost constant quasi-rectangular CV profiles exhibit minimal peak voltage shifts as the scan rate increases (Fig. 2d, e), signifying the low polarization of Fe||PTBA battery[47–49]. Besides, the PTBA positive electrode demonstrates highly reversible (dis)charge capacities and negligible voltage drops at various specific currents (Fig. 2f, Supplementary Fig. 17), highlighting the bipolar multi-redox structure

advantages and favorable electrochemical reversibility of PTBA. Fe|| PTBA battery delivers good cycling stability with 90.9% capacity retention after 5000 cycles at 0.5 A g$^{-1}$ (0.00182% per cycle, Fig. 2g), far surpassing other Fe||NTBA, Fe||TCB and Fe||TPP cells (Supplementary Figs. 18, 19). Even at a high specific current of 10 A g$^{-1}$, Fe||PTBA battery reaches an electrochemical lifespan of 60,000 cycles with 77.6% remaining capacity (0.00037% per cycle, Fig. 2h). SEM images and XPS analysis reveal that PTBA positive electrode maintains nearly identical morphologies and redox-active functional groups after prolonged cycling (Supplementary Figs. 20, 21), confirming its favorable structural stability and electrochemical reversibility.

Additionally, UV–Vis spectra analysis of Fe(OTF)$_2$/H$_2$O electrolyte after soaking cycled PTBA positive electrode shows no obvious absorption peak (Supplementary Fig. 22), indicating the resistance to dissolution of PTBA in aqueous electrolyte, thereby contributing to durable redox activity. Meanwhile, after experiencing repeated plating/stripping behavior during long-term cycling, the Fe negative electrode exhibits a relatively rough surface morphology (Supplementary Fig. 23). Therefore, the stable conjugated structure of the PTBA positive electrode is the reason for the durable operation of the Fe||PTBA battery. Furthermore, high-capacity-voltage PTBA positive electrode endows Fe||PTBA cell with a high specific energy of 217 Wh kg$^{-1}$ (based on the mass loading of PTBA in the positive electrode, 3 mg cm$^{-2}$, Supplementary Fig. 24). Fe||PTBA battery with

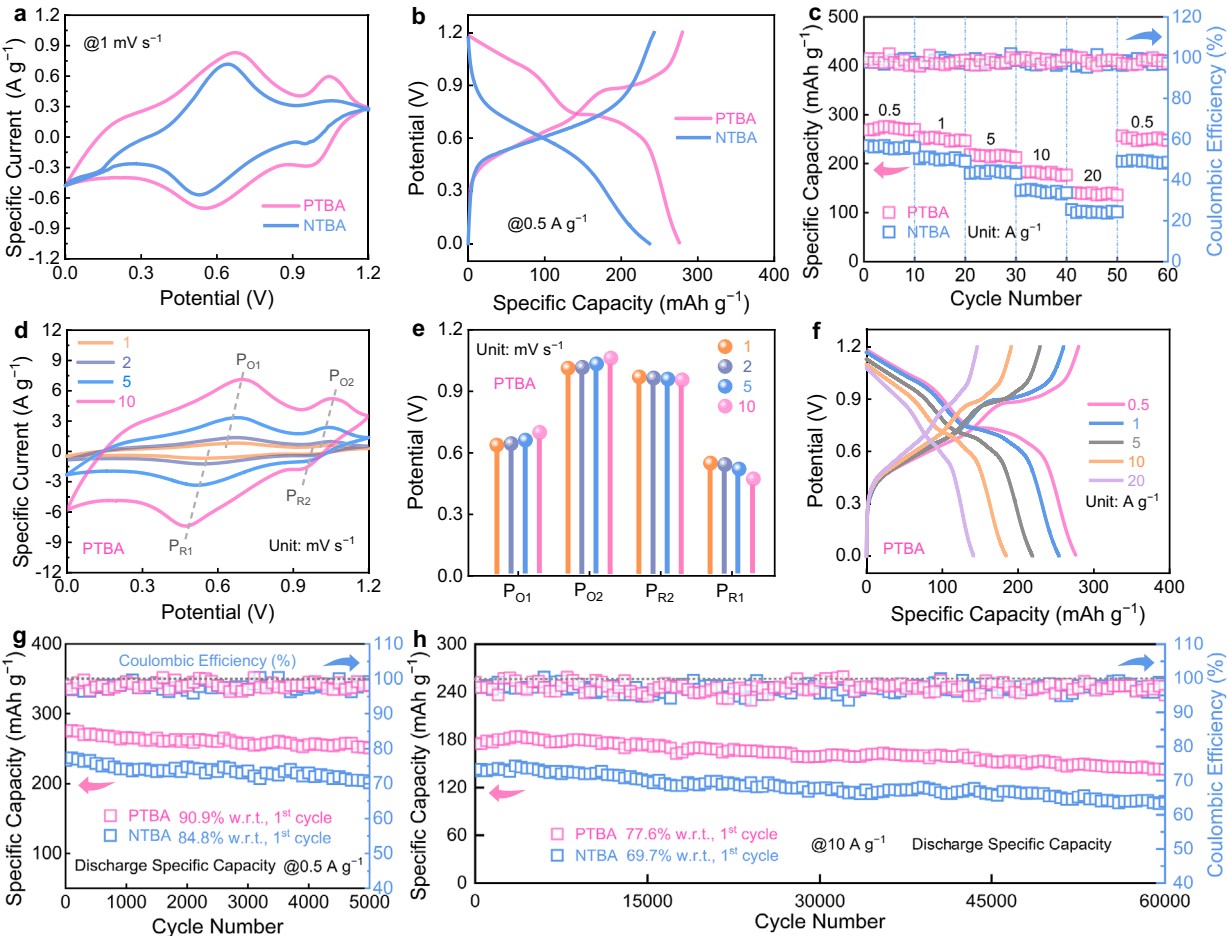

**Fig. 2 | Electrochemical performances.** All electrochemical tests were conducted at $25 \pm 0.5\,°C$ under ambient pressure. **a** CV profiles at $1\,mV\,s^{-1}$ within the potential range of $0-1.2\,V$, **b** GCD curves, **c** rate capacities of Fe||PTBA and Fe||NTBA batteries. **d** CV profiles at different scan rates of $1-10\,mV\,s^{-1}$, **e** peak voltages at various scan rates, **f** GCD curves of Fe||PTBA battery. **g** Cycling performance at low specific current. **h** Long-term cycling stability at high specific current.

$10.2\,mg\,cm^{-2}{}_{PTBA}$ still achieves high capacities of $229\,mAh\,g^{-1}$ at $1\,A\,g^{-1}$ and $147\,mAh\,g^{-1}$ at $10\,A\,g^{-1}$ (Supplementary Fig. 25), which is competitive compared to some recent relevant reports (120/53 mAh g$^{-1}$ at 1/ $10\,A\,g^{-1}$ under ~8 mg cm$^{-2}$)[15]. The desirable electrochemical performances highlight the electrochemical practical potential of phosphorus-engineered redox-active and stable organic materials towards better aqueous iron batteries. Besides, the assembled Fe || PTBA pouch-type cell, even at a high mass loading of $10.7\,mg\,cm^{-2}$ PTBA, demonstrates a high areal specific capacity of $2.25\,mAh\,cm^{-2}$ with 91.3% capacity retention after 1000 cycles at $1\,A\,g^{-1}$ (Supplementary Fig. 26), demonstrating its potential for practical applications. The comprehensively improved electrochemical metrics give the PTBA positive electrode the potential to compete with recently reported organics used for aqueous multivalent metal batteries (Supplementary Table 1).

## Structural evolution and mechanism analysis

The charge storage mechanism was investigated through systematic characterization using FT-IR spectroscopy and X-ray photoelectron spectroscopy (XPS). Five distinct (dis)charged states (A–E) in GCD curves of Fe||PTBA cell were analyzed to trace structural variation of PTBA positive electrodes during electrochemical reactions (Fig. 3a). The two-dimensional IR contour map reveals two characteristic absorption peaks at 1410 and 1685 cm$^{-1}$, corresponding to vibrational signals of −COO species and C=O species[50,51], maintaining relatively stable intensity during the initial discharge stage (states A to B, Fig. 3b).

Then these characteristic peaks gradually diminish as the discharge process deepens (states B to C), indicating n-type redox activity of carboxyl groups by coordinating with cations in the low potential area. Conversely, the C−P vibrational signal at 745 cm$^{-1}$ begins to appear from states A to B, and then remains constant during the following discharge (states B to C), confirming that phosphorus sites exhibit p-type redox behavior by removing anions in the high potential region. During the subsequent recharging process (states C to E), all absorption peaks reversibly return to the initial state A, which corresponds to the coupling reaction between carboxyl sites and Fe$^{2+}$ cations, as well as the decoordination process between phosphorus centers and OTF$^-$ anions.

To further probe the chemical bonding properties between Fe$^{2+}$/OTF$^-$ ions and carboxyl/phosphorus groups during (dis)charging, high-resolution XPS was employed to detect the C 1s signal of the PTBA positive electrode (Fig. 3c). At the initial state A, the deconvolution signal at 285.8 eV belonging to C=P$^+$ species, indicating p-type redox activity of C−P groups. Subsequently, during discharging (states A to B), the disappearance of C=P$^+$ species and the emergence of C−P bonds (states B to C) indicate the occurrence of a high-potential redox process involving the release of OTF$^-$ anions at phosphorus sites. In the continuous discharging process (states B to C), C=O species gradually decrease due to their participation in the reduction reaction at low potential, resulting in the formation of C−O···Fe moieties at 283.6 eV. Correspondingly, during the reverse charging process (states C to E), all species show reversible changes attributed to the removal of Fe$^{2+}$

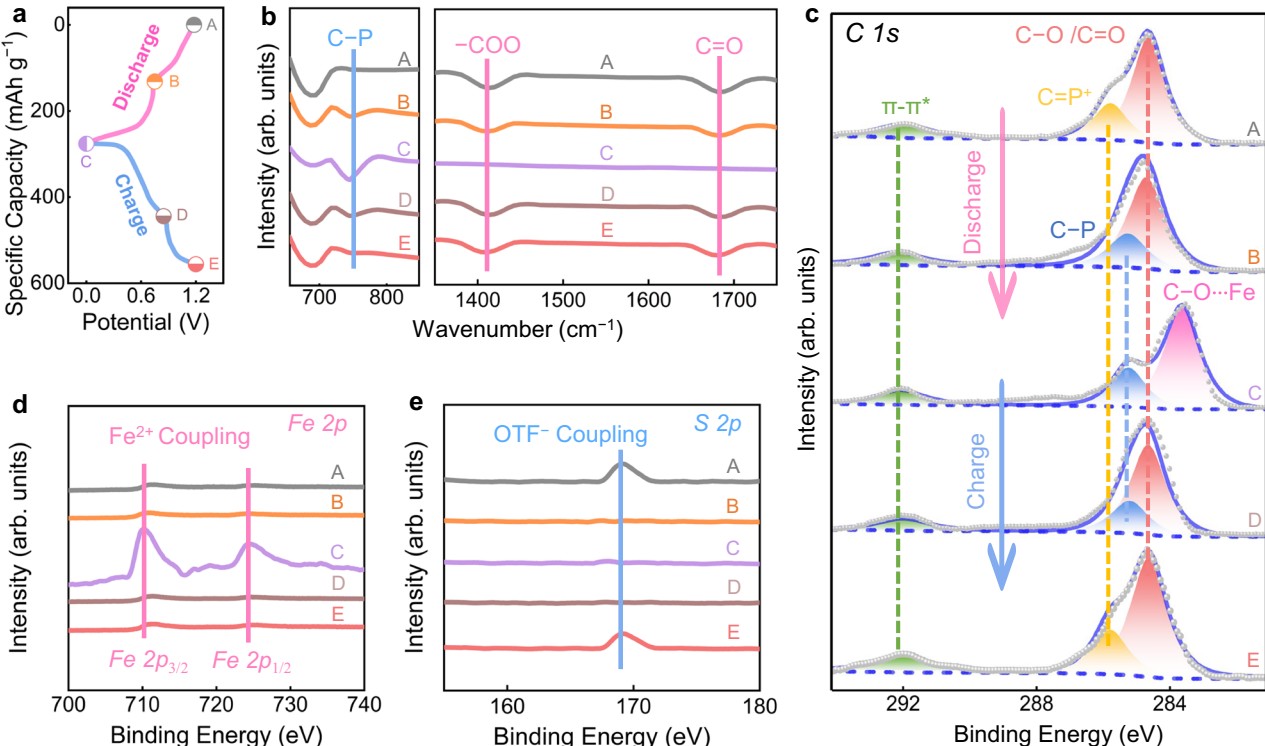

**Fig. 3 | Structural evolution of PTBA electrode during electrochemical reactions. a** A GCD curve. **b** Overview of FT-IR spectra. XPS spectra of **c** *C 1s*, **d** *Fe 2p*, and **e** *S 2p* corresponding to the selected voltages marked in the GCD curve. For ex situ spectrum analysis, cells were cycled at 0.5 A g$^{-1}$ for 3 cycles at 25 ± 0.5 °C under ambient pressure, and terminated at designated states of charge/discharge before disassembly.

cations from n-type carboxyl groups and the coordination between OTF$^-$ anions and p-type phosphorus sites. Furthermore, XPS spectra of O *1s* and P *2p* species exhibit evolution trends consistent with C *1s* spectra (Supplementary Fig. 27), collectively confirming the highly reversible redox activity of C=O and C−P groups in the PTBA positive electrode. Besides, the Fe *2p* and S *2p* spectra were employed to track the evolution of Fe$^{2+}$ cations and OTF$^-$ anions in the PTBA positive electrode during battery operation. Upon discharging (state B to C), the Fe *2p* signal gradually strengthens, reflecting the incorporation of Fe$^{2+}$ cations into the PTBA positive electrode. Then during charging (state C to D), Fe signal diminishes, demonstrating that Fe$^{2+}$ cations are reversibly removed to its initial state (Fig. 3d). In comparison, the S *2p* peak intensity, which correlates with OTF$^-$ transformation[52,53], exhibits an inverse trend relative to the Fe signal during (dis)charging (Fig. 3e), confirming the reversible coordination/decoordination of OTF$^-$. Overall, benefiting from the bipolar redox activity of n-type carboxyl groups and p-type phosphorus moieties in the PTBA electrode, hybrid anion–cation co-storage with the high utilization of active-sites is achieved.

To further eliminate interference from other ions and identify the active ions in Fe(OTF)$_2$/H$_2$O electrolyte, the electrochemical performance of Fe||PTBA battery was compared across various electrolytes. In the FeSO$_4$/H$_2$O electrolyte, the PTBA positive electrode shows moderate specific capacity (181 mAh g$^{-1}$ at 0.5 A g$^{-1}$) and cycling performance (84.6% capacity retention over 10000 cycles, Supplementary Fig. 15), demonstrating that Fe$^{2+}$ cations play a critical role in the whole charge storage process. Besides, the PTBA positive electrode in HOTF/H$_2$O electrolyte (similar pH to Fe(OTF)$_2$/H$_2$O electrolyte) exhibits fundamentally distinct electrochemical behavior with low specific capacity output (14 mAh g$^{-1}$, Supplementary Fig. 28), indicating that H$^+$ ions are hardly involved in the redox process of Fe||PTBA battery. The weak acidity and insufficient anions of the HOTF/H$_2$O electrolyte are unable to liberate the redox activity of p-type phosphorus groups,

leading to insignificant capacity storage. Therefore, Fe$^{2+}$ rather than H$^+$ ions are the active cations that dominate the energy storage of the PTBA positive electrode.

## Kinetics simulation and theoretical calculation

The kinetics behavior of Fe||PTBA battery in Fe(OTF)$_2$/H$_2$O electrolyte was further assessed through Dunn's analysis[54,55]. By applying the power-law relationship of $i = kv^b$ between the peak current ($i$) and scan rate ($v$)[56–58], all four cathodic/anodic peaks of CV profiles exhibit high b-values approaching 1 (Fig. 4a), suggesting that the PTBA positive electrode exhibits rapid surface-dominant capacitive behavior (80.6−95.9% contribution) along with negligible diffusion-limited process across all scan rates (Fig. 4b, c). Temperature-dependent electrochemical impedance spectroscopy (EIS) measurements of the PTBA positive electrode reveal characteristic Nyquist plots consisting of semicircular high-frequency regions and linear low-frequency tails[59–61]. Analysis of the Nyquist plots using the equivalent circuit model (Fig. 4d, e) exhibits low interfacial $R_{ct}$ values (12.5 ~ 42.0 Ω) for PTBA during Fe$^{2+}$/OTF$^-$ coordination (Supplementary Tables 2, 3). Arrhenius analysis of temperature-dependent electrochemical behavior[62–64], demonstrating that lower interfacial activation energies ($E_a$) of PTBA (0.24 eV) compared to NTBA (0.29 eV, Fig. 4f), implying the fast ion transport kinetics and low energy barriers for charge storage of the PTBA positive electrode. Overall, Fe||PTBA battery demonstrates enhanced electrochemical performance metrics (*e.g.*, specific capacity, rate performance, specific energy, and cycle life), which stems from the multiple bipolar redox centers, extended π-conjugated skeleton and low-steric-hindrance configuration of the PTBA positive electrode.

Given the negligible specific capacity contributions from acetylene black (8 mAh g$^{-1}$, Supplementary Fig. 16), the real specific capacity of the PTBA positive electrode of 268 mAh g$^{-1}$ is calculated to be approximately 4e$^-$ redox reactions, corresponding to an actual

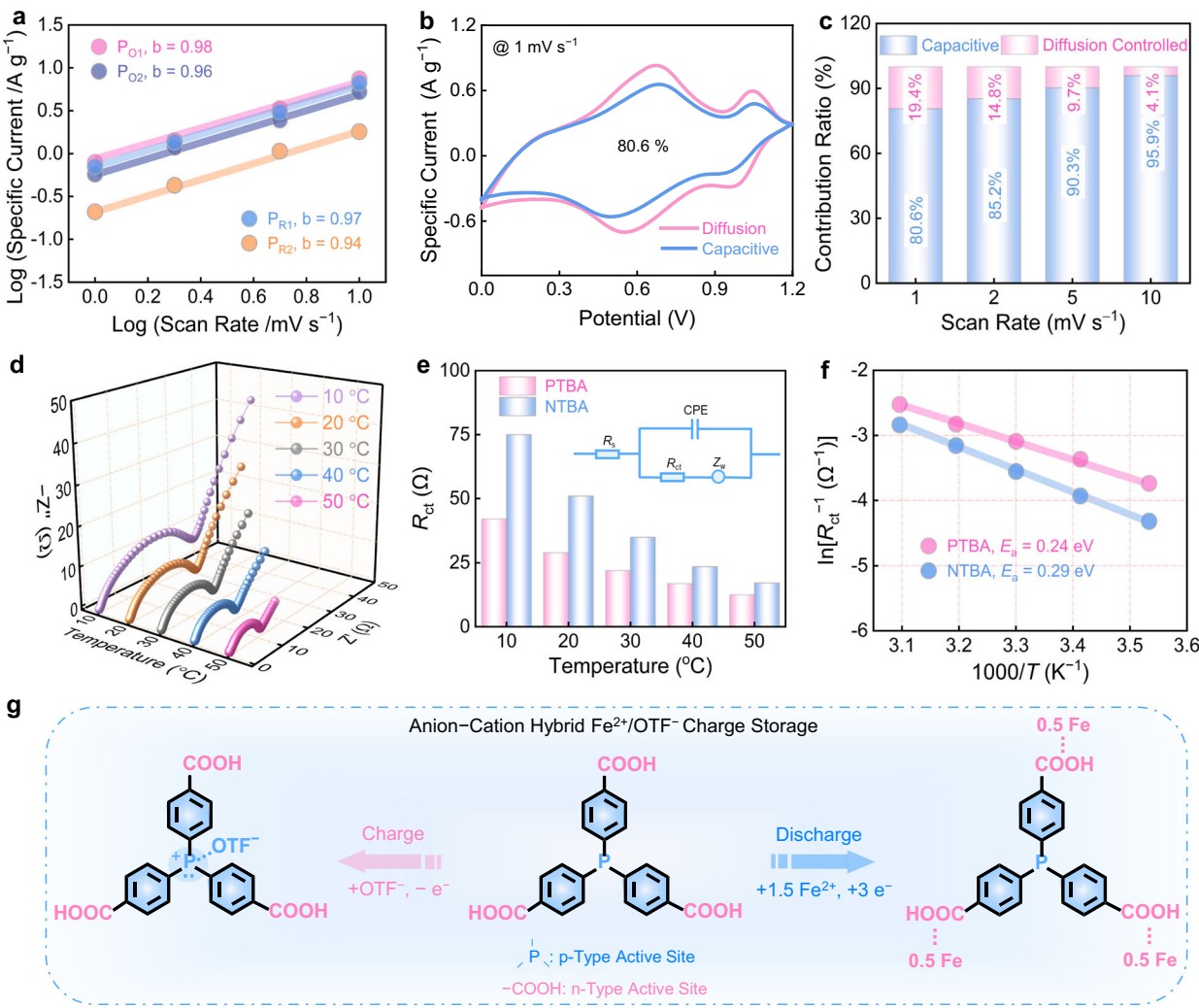

**Fig. 4 | Kinetics simulation of PTBA. a** Evaluated b values. **b** Capacitive contribution at 1 mV s$^{-1}$. **c** Capacitive contribution ratios at different scan rates. **d** EIS results of the PTBA electrode at various temperatures. **e** Corresponding $R_{ct}$ values of PTBA and NTBA electrodes. Inset: the equivalent circuit of Nyquist plots ($R_s$: equivalent series resistance; $R_{ct}$: charge transfer resistance; $Z_w$: Warburg impedance; CPE: constant phase element). **f** Calculated $E_a$ values. **g** Schematic diagram of anion-cation hybrid storage in the PTBA electrode during redox processes.

utilization efficiency of 98.5% for n–p fused carboxyl/phosphorus motifs. In a word, these experimental results shed light on the fact that Fe$^{2+}$/OTF$^-$ co-coordination with carboxyl/phosphorus sites of PTBA positive electrode via a successive two-step four-electron bipolar redox mechanism to achieve fast and stable hybrid storage energy (Fig. 4g).

Density functional theory (DFT) simulations provided further insights into the structural changes and electron configuration of PTBA positive electrodes during battery operation. The optimized geometry of PTBA indicates the charges of −COOH and C−P groups are approximately −0.476 and −0.651 a.u., respectively, highlighting their strong coordination ability (Fig. 5a). The continuous ELF-π isosurfaces across PTBA framework indicate efficient π-electron delocalization and π-aromaticity along the extended conjugation structure (Fig. 5b). In the reduced density gradient (RDG) plots (Fig. 5c, Supplementary Data 2), the green signature (−0.02 ~ 0.00 a.u.) and blue spike (−0.04 ~ −0.02 a.u.) reveal the strong hydrogen bonds and π-π stacking interactions between PTBA molecules. The anisotropy of the induced current density (ACID) analysis demonstrates persistent current flowing throughout the entire PTBA backbone, confirming its global π-aromatic nature (Supplementary Fig. 29). During charging, an OTF$^-$ anion binds to the phosphorus

site of pristine PTBA with a calculated binding energy ($\Delta E_1$) of −1.23 eV, facilitating the formation of the anion-doped PTBA-OTF$^-$ complex (state I). Subsequently, during discharging, Fe$^{2+}$ cations coordinate with carboxyl sites of PTBA, where the associated $\Delta E_2$ of −0.43 eV further promotes the generation of cation-coordinated PTBA-1.5Fe$^{2+}$ products (state II, Fig. 5d, Supplementary Data 3). Consequently, the synergistic anion-cation binding configuration enables rapid and stable hybrid energy storage in PTBA during the operation of IOBs. This is evidenced by MEP distribution, which confirms the structural stability of both PTBA-OTF$^-$ and PTBA-1.5Fe$^{2+}$ (Fig. 5e).

The charge-density difference isosurface analysis was performed to elucidate the bonding characteristics of the anion/cation-coordinated PTBA complexes (Fig. 5f). The observed charge redistribution, characterized by charge depletion around charge carriers and charge accumulation near −COOH/C−P moieties, reveals strong redox interactions that facilitate structural stabilization. The quantitative analysis of Bader charge further reveals significant charge transfer between charge carriers and bipolar redox-active sites (State I: 0.53 e; State II: 0.27 e), implying the high redox activity of electron-donating n-type carboxyl motifs and electron-accepting p-type phosphorus moieties. The alternating cation-anion storage (Fe$^{2+}$/OTF$^-$) facilitates full

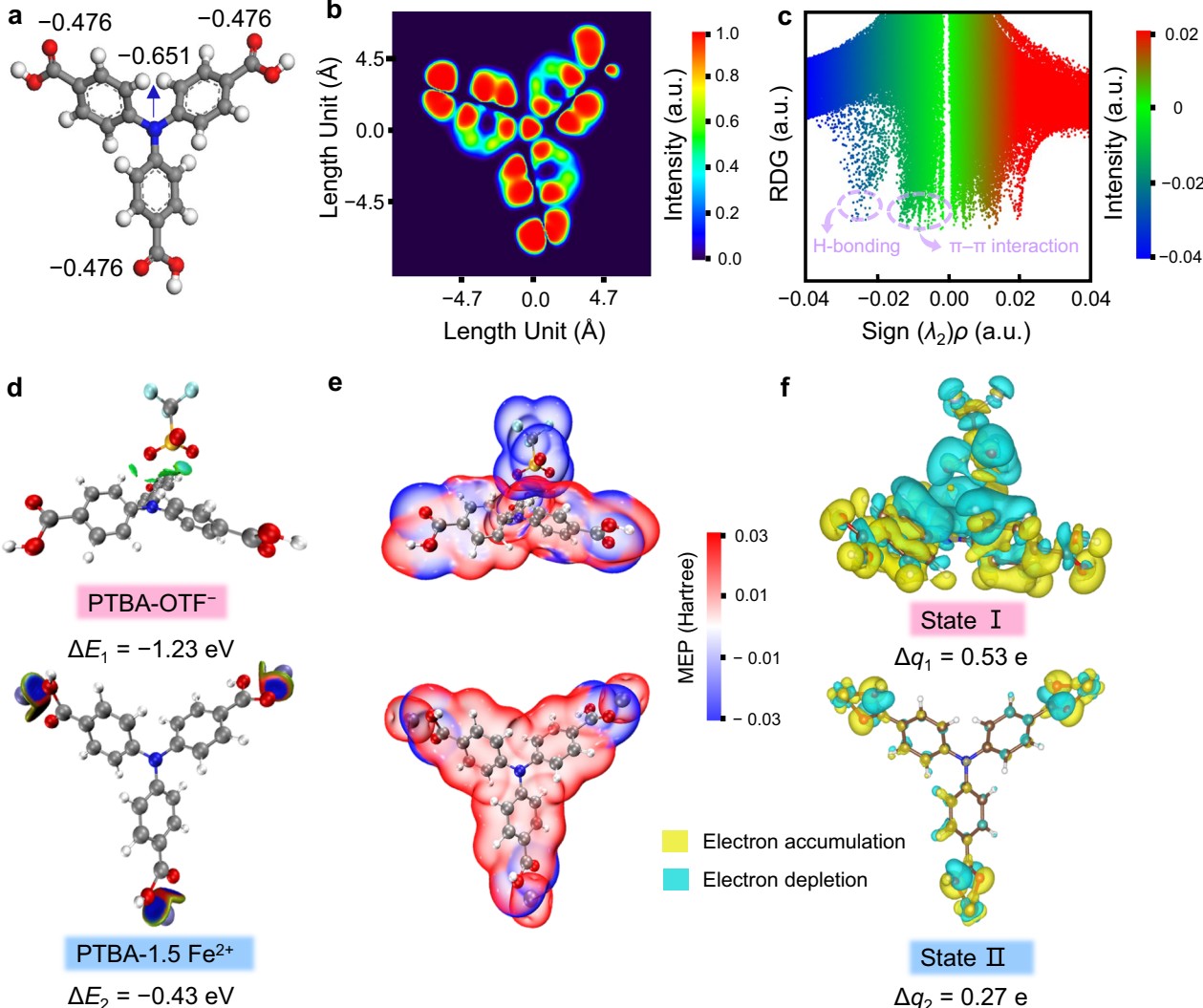

**Fig. 5 | Theoretical simulations of BTPA. a** Charge sum of active sites. **b** ELF-π image and **c** RDG plots of PTBA. **d** Binding energy, **e** structural changes and MEP distribution, **f** charge density difference analysis of PTBA-OTF⁻ and PTBA-1.5Fe²⁺. Colors of elements: C, gray; O, red; H, white; P, blue; N, yellow.

utilization of the multi-redox-active carboxyl/phosphorus sites in PTBA, leading to enhanced electrochemical performance.

In conclusion, the phosphorus redox activity is activated in PTBA with a P-extended conjugated structure. By modulating the electron-donating P and tert-N groups, 4,4′,4″-Phosphanetriyl-tribenzoic acid (PTBA) and 4,4′,4″-nitrilotribenzoic acid (NTBA) are designed, which show various charge distributions and energy levels. With the decline of the electronegativity and energy barrier (N > P), PTBA shows stronger $Fe^{2+}$ coordination with carboxyl sites, and brings the $CF_3SO_3^-$ proximity to P centers. It ensures high redox activity of carboxyl/phosphorus sites with a low activation energy (0.24 *vs.* 0.29 eV for NTBA). Furthermore, PTBA shows the low energy gap compared to its individual carboxyl or P-containing counterparts, thus giving high utilization of carboxyl/P motifs (98.5%) and enhanced redox voltage (0.8 V). The phosphorus-linked π-conjugated aromatic framework of PTBA further addresses the dissolution issue, enabling stable 4 e⁻ $Fe^{2+}/CF_3SO_3^-$ storage. The assembled Fe‖PTBA battery achieves promising performance metrics, including high specific capacity, rate capability, specific energy and cycling stability, showing good potential for iron-based energy storage. This work broadens the design horizons of phosphorus-engineered organic materials for aqueous iron batteries.

## Methods

### Materials

All reagents were commercially available and directly used without further purification. 4,4′,4″-phosphanetriyltribenzoic acid (PTBA, 99%), 4,4′,4″-nitrilotribenzoic acid (NTBA, 99%), 1,3,5-tris(4-carboxyphenyl)benzene (TCB, 99%), and triphenylphosphorus (TPP, 99%) were purchased from Jilin Chinese Academy of Sciences-Yanshen Technology Co., Ltd. Polyvinylidene difluoride (PVDF, $M_w$ = 1200000 Da, 99.5%), acetylene black (particle size: 40–50 nm, 99.5%), separator (Whatman CF/D, thickness: 675 μm, lateral dimension: 90 mm, porosity: 90 ± 2%, pore size: 2.7 μm, one separator for one battery), coin cell components (CR2032, 316 stainless steel, positive electrode case: 18 × 2.91 mm, negative electrode case: 16 × 2.77 mm, spacer: 15.8 × 1 mm, spring: 15.6 × 1.1 × 0.25 mm) were purchased from the Guangdong Canrd New Energy Technology Co., Ltd. Carbon felt (thickness: 0.05 mm) was purchased from Wuhu Eryi Material Technology Co., Ltd. Iron foil negative electrode (>99.99% purity, prepared directly prior to cell assembly), deionized water ($H_2O$, 99.99%), iron(II) trifluoromethanesulfonate (Fe(OTF)₂, >97.0%), ferrous sulfate heptahydrate (FeSO₄·7H₂O, >99.0%), trifluoromethanesulfonic acid (HOTF, 99%) and N, N-dimethylformamide (DMF, 99.8%) were purchased from Titan Co., Ltd. The electrolyte is freshly prepared and used. After stirring the electrolyte evenly with a

glass rod, it is placed in a volumetric flask and transferred for application using a polypropylene pipette. The electrolyte is then placed at room temperature and atmospheric pressure.

## Material characterizations

The morphology and elemental composition of organics were examined using scanning electron microscopy (SEM, Hitachi S-4800) coupled with an X-ray energy dispersive spectroscopy (EDS) instrument. Fourier-transformed infrared spectra (FT-IR) were carried out via a Thermo Nicolet NEXUS spectrometer. X-ray diffraction (XRD) pattern was monitored by using a Bruker D8 Advance powder diffractometer (Cu Kα radiation source). The ultraviolet visible (UV-Vis) spectra were recorded using a UV-Vis spectrometer (JASCO V-750). The electrical conductivity of PTBA and NTBA positive electrodes was tested at 4 different points on the electrode surface with an RTS-8 four-point probe. An X-ray photoelectron spectrometer (XPS, AXIS Ultra DLD) was applied to study the chemical composition. To explore the dynamic evolution of organic positive electrodes during the (dis) charging process, PTBA positive electrodes were collected by disassembling aqueous Fe-organic batteries at specific (dis)charging voltages at $25 \pm 0.5\,°C$ under ambient pressure. Various ex-situ spectroscopic characterizations of PTBA positive electrodes, including FT-IR, XPS and SEM, were investigated.

## Fabrication of working electrodes

The working electrodes were prepared by mixing 60 wt. % PTBA (active material), 30 wt. % acetylene black (conductive additive), and 10 wt. % PVDF (binder) by using DMF solvent in a mortar with pestle. The wet slurry was stirred for 2 h to ensure homogeneity and then cast onto a carbon felt current collector using an automatic coating machine (MSK-AFA-III). The coated electrodes were dried at 80 °C for 12 h under vacuum. Subsequently, the electrodes were punched into 14 mm disks using a manual disc cutter (MSK-T10) and further dried at 80 °C overnight under vacuum prior to cell assembly. The areal mass loading of the active material was approximately 3 mg cm$^{-2}$ (thickness: 90 μm). The working electrode surface was wetted by using a polypropylene pipette to transfer 30 μL Fe(OTF)$_2$ electrolyte.

## Cell assembly

To construct aqueous iron-organic batteries (IOBs), PTBA (or NTBA, TCB, TPP) positive electrode (PTBA, 3 mg cm$^{-2}$, single-side coated, areal capacity: 0.83 mAh cm$^{-2}$), 1 mol L$^{-1}$ aqueous Fe(OTF)$_2$ electrolyte (135 μL), Fe metal negative electrode (>99.99% purity, density: 7.87 g cm$^{-3}$, 50 μm in thickness, 1.6 cm in diameter, capacity ratio between Fe negative and PTBA positive electrodes (N/P) is 60:1, and glass fiber separator (Whatman, 18 mm) were packaged into 2032 coin-type cells. The initial cut-off potential of the cell is -0.7 V for a newly-assembled Fe||PTBA coin cell. For the assembly of single-layer pouch cell, PTBA positive electrode (with carbon felt current collector, mass loading: 10.7 mg cm$^{-2}$, 8 × 8 cm$^2$), 1 mol L$^{-1}$ aqueous Fe(OTF)$_2$ electrolyte (3 mL), Fe metal negative electrode (>99.99% purity, 50 μm in thickness, 8 × 8 cm$^2$, capacity ratio between Fe negative and PTBA positive electrodes (N/P) is 13:1, and glass fiber separator (Whatman, 9 × 9 cm$^2$) were packaged. Three identical Fe||PTBA cells were tested for a single electrochemical experiment. The data presented in the manuscript's plots belong to only a specific selected cell.

## Electrochemical tests

Galvanostatic charge/discharge (GCD) profiles, rate performance and cyclic stability of IOBs were tested on a Neware battery test system (CT-4008Tn-5V10mA-164, Shenzhen, China) within the potential range of 0−1.2 V. Cyclic voltammetry (CV) and electrochemical impedance spectroscopy (EIS) tests were performed on an electrochemical workstation (CHI660E). For EIS measurements, a potentiostatic perturbation with an amplitude of 5 mV was applied over a frequency range of 0.01 Hz to 100 kHz (with a data density of 12 points per decade). Prior to each EIS measurement, the cells were held at open-circuit potential for 1000 s to establish a quasi-stationary state. All electrochemical tests were conducted at $25 \pm 0.5\,°C$ under ambient pressure. The specific capacity ($C_m$, mAh g$^{-1}$) was estimated from GCD curves based on the following equations:

$$C_m = \frac{I \times \Delta t}{m} \tag{1}$$

where $I$, $\Delta t$, $m$ represents the specific current (A g$^{-1}$), discharging time (s) and the effective mass loading of active materials in positive electrodes, respectively. When considering the molecular weight of 394.3 g mol$^{-1}$ and four-electron transfer during the redox process, the theoretical capacity is calculated to be 271.9 mAh g$^{-1}$ for the PTBA electrode. Similarly, the theoretical capacity of NTBA is 284.1 mAh g$^{-1}$. The Coulombic efficiency (CE) of PTBA and NTBA positive electrodes was calculated as the ratio of discharge capacity divided by the charge capacity in the preceding charge cycle.

The specific energy ($E$, Wh kg$^{-1}$) and specific power ($P$, W kg$^{-1}$) of IOBs were calculated according to the following equations:

$$E = C_m \times \Delta V \tag{2}$$

$$P = \frac{E}{1000 \times \Delta t} \tag{3}$$

where $\Delta V$ is the voltage window. The electrochemical metrics, including specific capacities and specific energy/power, were estimated based on the effective mass loading of active materials in the positive electrode.

## Density Functional Theory (DFT) calculation

The theoretical calculations were performed with the Gaussian 16 software package. Geometrical optimizations were conducted at the B3LYP-D3/ def2-SVP level of theory. The negative molecular electrostatic potentials (MEP) in the red region represent the electrophilic nature, while the positive MEP in the blue region represents the nucleophilic nature. The π-electron localization function (ELF-π) was evaluated via the Multiwfn 3.8 programs. The molecular orbital levels of organics, in terms of the highest occupied molecular orbital (HOMO) and the lowest unoccupied molecular orbital (LUMO), as well as the charge population sum of PTBA, were studied at the B3LYP-D3/ TZVP level of theory. The binding energy between charge carriers and organics was calculated using the Vienna Ab initio Simulation Package (VASP) with the projector augmented wave (PAW) method. The exchange-functional was treated using the generalized gradient approximation (GGA) with Perdew-Burke-Emzerhof (PBE) functional. The charge density differences were recognized by the VASPKIT code. The charge density differences between charge carriers and organic models were simulated, and the charge transfer level between both was calculated by Bader charge analysis. The anisotropy of the current-induced density (ACID) analysis was realized by the ACID code based on the output file of Gaussian 16, and the maps were finally generated by POV-Ray render.

## Optical energy gap

The optical energy gaps ($E_g$, eV) of organics were calculated from UV-Vis spectra based on the following equations:

$$\alpha \propto \frac{(h\upsilon - E_g)^{1/2}}{h\upsilon} \tag{4}$$

$$h\upsilon = 1280/\lambda \tag{5}$$

where $\alpha$ is the optical absorption coefficient, $h\nu$ is the photon energy, $\lambda$ is the wavelength.

## Activation energy

The activation energy ($E_a$, eV) for the charge transfer process was calculated by the Arrhenius equation:

$$R_{ct}^{-1} = A \exp(-E_a/RT) \tag{6}$$

where $R_{ct}$ refers to the charge transfer resistance ($\Omega$), $R$ is the gas constant (8.314 J mol$^{-1}$ K$^{-1}$), $T$ is the experimental temperature ($K$), and $k$ is a constant.

## Charge storage kinetics

The charge storage kinetics of IOBs were evaluated by the following equation:

$$i = k\nu^b \tag{7}$$

where $k$ and b refer to constants, $i$ and $\nu$ are specific current and scanning rate, respectively. When the power exponent b value is close to 0.5, it identifies a sluggish diffusion-controlled process, while a b-value of 1.0 suggests a rapid surface-dominated redox reaction process.

The Dunn's method was applied to analyze the capacitive contribution from the rapid surface redox capacitive process and the diffusion-limited process. Quantitative capacitive contribution can be calculated according to the equation:

$$i = k_1\nu + k_2\nu^{1/2} \tag{8}$$

where $k_1$ and $k_2$ are constants, $k_1\nu$ and $k_2\nu^{1/2}$ are the specific current contributed from fast-capacitive process and diffusion-controlled process, respectively.

## Redox electron transfer number

The theoretical capacity ($C_m$, mAh g$^{-1}$) of an organic positive electrode can be calculated based on the following form:

$$C_m = \frac{n \times F}{3.6 \times M} \tag{9}$$

where $F$ is a constant (96485 C mol$^{-1}$), $M$ is the molar mass of an organic material (g mol$^{-1}$).

## Data availability

The data generated in this study are provided in the Source Data file. Source data are provided with this paper.

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

## Acknowledgments

This work is financially supported by the National Natural Science Foundation of China (NO. 22272118, M.L.; NO. 22172111, L.G.; NO. 22309134, Z.S. and NO. 22502144, Y.Z.), the Shanghai Rising-Star Program (23YF1449200, Z.S.), the Postdoctoral Fellowship Program of CPSF (GZC20250749, Y.Z.), the Zhejiang Provincial Science and Technology Project (NO. 2022C01182, Y.L.), and the Fundamental Research Funds for the Central Universities (Z.S.).

## Author contributions

Y.Z. conceived the idea and designed the project. M.L. supervised the experiments and edited the paper. Z.S., L.G., Y.L., and P.L. performed the data processing and analysis. Q.H. performed a computational simulation. Y.Z., Z.S., and M.L. contributed to the manuscript review. All authors discussed the results and contributed to the completion of the manuscript.

## Competing interests

The authors declare no competing interests.
