## [Transparent Peer Review file · Nature Communications]

Phosphorus-Activated Carboxyl Small Molecule Positive Electrode for High Specific Capacity and Long-Life Iron-Organic Batteries

Corresponding Author: Professor Mingxian Liu

Version 0:

Reviewer comments:

Reviewer #1

(Remarks to the Author)

In this work, the authors activate novel phosphorus redox activity in a carboxyl-based small molecule. While it provides a systematic investigation of phosphorus-containing organic cathodes for iron-ion batteries, it exhibits limited breakthrough innovation, particularly in terms of energy storage mechanisms and performance under practical conditions. Therefore, this manuscript is not recommended for publication in a high-level journal such as Nature Communications. The main concerns are summarized as follows:

1. While this study presents a novel structural design with n-p bipolar characteristics for aqueous Fe-ion batteries, it should be noted that the concept of n-p bipolar design itself is not new and has been previously documented in organic electrode materials. The work primarily transfers this established concept to a different material system, without introducing substantial innovation in the reaction mechanisms at active sites. In essence, it adopts existing principles while implementing them in a new material context.
2. Currently, research on organic electrode materials should be oriented toward practical applications, such as evaluating their performance under practical conditions (e.g., tests under high mass loading and high areal capacity, validation in pouch cells). However, this work is limited to performance tests under low mass loading, which cannot adequately assess the practical application prospects of the materials. Therefore, this study has not achieved a breakthrough in performance under practical conditions.
3. The authors should perform electrochemical performance tests under practical conditions (such as high mass loading, high areal capacity, and pouch cell configuration) to better evaluate the practical potential of the material.
4. In Figure 1f, acetylene black conductive agent was added during the electronic conductivity test. However, electronic conductivity measurements should evaluate the intrinsic properties of the pure material without incorporating any additives.
5. In Figure 3c, the original data should be included alongside the XPS spectra.
6. In Figure S12, the schematic is inaccurate. Fe^{2+} intercalation can also occur in the PTBA cathode.
7. In Figure S13, it is reasonable that the PTBA cathode containing 30% acetylene black additive exhibits superior capacities compared to the 20% counterpart, which can be attributed to improved electronic conductivity. However, the fact that the PTBA cathode with 30% acetylene black additive delivers higher capacity than the 40% counterpart requires further explanation from the authors.

Reviewer #2

(Remarks to the Author)

Comment: The manuscript entitled "Phosphorus-Activated Carboxyl Small Molecule Cathode for High-Capacity and Long-Life Iron-Organic Batteries" submitted by Zhang et al. presents the development of two organic molecules with bipolar features as cathode materials for aqueous iron-ion batteries. By modulating the p-type electron-donating phosphorus and tert-N motifs, 4,4',4''-phosphanetriyltribenzoic acid (PTBA) exhibits tailored charge distributions and energy levels, endowing the Fe||PTBA battery with state-of-the-art capacity (276 mAh g⁻¹) and outstanding cycling stability (60,000 cycles) among reported cathode materials in IIBs. The authors investigated the Fe²⁺ ion storage pathways in the PTBA electrode through XPS, FTIR, and DFT calculations. However, the following questions need to be addressed.

- 1 What are the comparative advantages of iron-ion batteries over zinc-ion batteries in aqueous systems, based on the latest advancements in the field? These should be clarified in the Introduction section.
- 2 More evidence is required to demonstrate the contribution of carboxyl groups to the capacity of IOBs.
- 3 The GCD curves of Fe||TCB and Fe||TPP batteries should be provided.
- 4 The electrical conductivity of the electrode improves with increasing conductive carbon content, yet an unexpected reduction in capacity is observed, which contradicts prior findings. What underlying mechanisms could explain this phenomenon?
- 5 The energy storage mechanism is investigated by the ex-situ XPS. The authors gave the variation of C 1s XPS spectra during the charging/discharging process. However, the O 1s and P 2p spectra XPS spectra of the PTBA electrode are not given. Therefore, the O and P XPS spectra should be deconvoluted and analyzed to prove the reversible reaction in the PTBA structure during the charging/discharging process.
- 6 Has the morphology of the PTBA material changed after long-term cycling?
- 7 It should be verified whether hydrogen ions participate in the energy storage process during the reaction.

Reviewer #3

(Remarks to the Author)

The authors demonstrated phosphorus redox activity in a carboxyl small molecule, through designing two bipolar carboxyl organic small molecules, 4,4',4"-phosphanetriyltribenzoic acid (PTBA) and 4,4',4"-nitrilotribenzoic acid (NTBA) with tuned charge distributions and energy levels by incorporating the electron-donating p-type phosphorus and tert-N motifs as cathode materials for iron-organic batteries (IOBs). This study not only discovered new-type phosphorus redox chemistry in carboxyl cathode, but also solved the capacity and durability limitations of prevailing polyaniline cathodes, including state-of-the-art capacity (276 mAh g⁻¹) and outstanding cycling stability (60,000 cycles). This manuscript is of importance with well-organized analysis and discussion, and also highlights the potential of phosphorus-engineered redox-active and stable organic materials toward better IOBs. Thus, I recommend its publication in Nature Communications after a revision.

- (1) What are the theoretical capacities of PTBA and NTBA? Is the ultrahigh utilization rate of 98.5% the actual utilization efficiency?
- (2) Why choose Fe(OTf)₂/H₂O solution as the electrolyte? The reason behind this behavior should be explained. Is PTBA cathode also applicable to the cheaper FeSO₄/H₂O electrolyte?
- (3) Given the inherent irreversibility of iron plating/stripping processes, which are invariably accompanied by parasitic reactions. So how can its outstanding cycle stability (60,000 cycles, Fig. 2h) be explained?
- (4) Stable electrochemical performance in high-mass-loading organic electrodes (>10 mg cm⁻²) is critical for practical battery applications. High-quality cycling data for PTBA cathodes under these conditions would be particularly compelling, as it could pave the way for iron-organic batteries with improved large-current durability.
- (5) Given the claimed outstanding stability of PTBA cathode, so it is recommended to supplement the comparative XPS analysis before and after prolonged cycling.
- (6) Essential battery fabrication parameters require clarification, particularly regarding Fe anode diameter, separator specifications and electrolyte content. Additionally, the initial open-circuit voltage measurement for a newly-assembled coin cell should be reported.

Version 1:

Reviewer comments:

Reviewer #1

(Remarks to the Author)

The authors have adequately addressed all concerns raised. The additional data under high mass loading and in a pouch cell configuration convincingly demonstrate the practical potential of the material. The revisions have strengthened the manuscript, clearly highlighting the novelty of the "phosphorus-activated redox chemistry" concept with robust mechanistic support. I believe this work has reached the standard for publication in Nature Communications and recommend its acceptance.

Reviewer #2

(Remarks to the Author)

Since the authors have clarified almost all the issues concerned, I suggest the acceptance of this paper for publication in this journal in the present form.

Reviewer #3

(Remarks to the Author)

My concerns have been properly addressed. Current manuscript can be considered acceptance.

Point-to-Point Response to Reviewer Comments

Response to the Comments of Reviewer #1:

We would like to express our sincere thanks to the expert reviewer for the constructive and insightful comments which significantly contributed to improving the quality of the manuscript. Please find below a detailed response to each of the comments.

Reviewer #1: In this work, the authors activate novel phosphorus redox activity in a carboxyl-based small molecule. While it provides a systematic investigation of phosphorus-containing organic cathodes for iron-ion batteries, it exhibits limited breakthrough innovation, particularly in terms of energy storage mechanisms and performance under practical conditions. Therefore, this manuscript is not recommended for publication in a high-level journal such as Nature Communications. The main concerns are summarized as follows:

1. While this study presents a novel structural design with n-p bipolar characteristics for aqueous Fe-ion batteries, it should be noted that the concept of n-p bipolar design itself is not new and has been previously documented in organic electrode materials. The work primarily transfers this established concept to a different material system, without introducing substantial innovation in the reaction mechanisms at active sites. In essence, it adopts existing principles while implementing them in a new material context.

Response: Thank you for your insightful comments. In this work, our proposed new concept is phosphorus-activated redox chemistry in a carboxyl small molecule as powerful electrochemical protocols for better aqueous iron-organic batteries (IOBs), rather than the well-established n-p bipolar design, just as commented by Reviewer 3: "This study not only discovered new-type phosphorus redox chemistry in carboxyl cathode, but also solved the capacity and durability limitations of prevailing polyaniline cathodes, including state-of-the-art capacity (276 mAh g^{-1}) and outstanding cycling stability (60,000 cycles)."

We further studied the phosphorus-activated redox chemistry with supplementary electrochemical experiments of TCB cathode (only containing carboxyl groups) and TPP cathode (only containing phosphorus) in Fig. S18, and added related discussion behind Fig. S18 in the revised Supplementary Information: TCB (containing carboxyl) and TPP cathodes (containing phosphorus) exhibit relatively low capacity of 121 mAh g^{-1} (66% carboxyl utilization) and 59 mAh

g^{-1} (57.7% phosphorus utilization, Fig. S18), respectively, which can be further boosted to 276 mAh g^{-1} for PTBA cathode (containing carboxyl/phosphorus sites) with 98.5% utilization (Fig. 2b). This is because that PTBA cathode with P-extended conjugated structure liberates the lowest energy gap (2.28 eV) compared with TCB/TPP cathode (2.71/3.16 eV, Fig. S9), which significantly increases the utilization of carboxyl redox sites. In addition, despite the 1 e^- capacity contribution of P site (vs. total 4 e^- charge storage), it boosts the average discharge voltage of PTBA to 0.8 V (Fig. 2b), significantly higher than TCB cathode (0.5 V). These results highlight the importance of P group in PTBA for promoting high voltage and high carboxyl utilization for enhanced capacity storage.

Fig. S18. GCD curves of (a) Fe||TPP and (b) Fe||TCB batteries in 1M Fe(OTf)₂/H₂O.

To date, there have been a few preliminary investigations on organic cathode materials for IOBs. The prevailing polyaniline cathodes for IOBs have demonstrated promising electrochemical redox properties, but their twisted polymeric chains and unstable structural linkages often result in disordered stacking structures, resulting in low utilization of redox-active sites (<80%) with reduced capacity (<250 mAh g^{-1}) and limited cycling durability. Thus, more attempts are still needed to unlock the electrochemical limitations of IOBs by strategic structural engineering.

Different from previously reported works, we first demonstrate the redox activity of phosphorus in a carboxyl small molecule cathode with easy-accessible redox sites and stable structures for propelling high-capacity and durable aqueous IOBs. We would like to emphasize the novelty and importance of our work as follows:

(1) Two organic molecules of 4,4',4''-phosphanetriyltribenzoic acid (PTBA) and 4,4',4''-nitrilotribenzoic acid (NTBA) are designed *via* modulating the electron-donating P and tert-N motifs, which show tuned charge distributions and energy levels.

(2) With the decrease of the electronegativity and energy barrier ($N > P$), PTBA exhibits stronger Fe^{2+} coordination with carboxyl sites, and brings closed CF_3SO_3^- proximity to P centers. This feature ensures high redox activity of carboxyl/phosphorus sites with a low activation energy (0.24 vs. 0.29 eV for NTBA).

(3) PTBA with P-extended conjugated structure achieves the lowest energy gap (2.28 eV) compared to its individual carboxyl or P-containing counterparts (2.71/3.16 eV), thereby enabling ultrahigh utilization of carboxyl/P motifs (98.5%) and enhanced redox voltage (0.8 V).

(4) A stable $4 e^- \text{Fe}^{2+}/\text{CF}_3\text{SO}_3^-$ storage is thus activated in P-linked π -extended PTBA cathode, endowing Fe||PTBA battery with state-of-the-art capacity (276 mAh g^{-1}) and cycling stability (60,000 cycles) among all reported cathode materials in iron batteries.

To give a better understanding of the new concept of phosphorus-activated redox chemistry for general readers, we have appropriately revised key sections of the manuscript including the Abstract, Introduction, and Conclusion.

2. Currently, research on organic electrode materials should be oriented toward practical applications, such as evaluating their performance under practical conditions (e.g., tests under high mass loading and high areal capacity, validation in pouch cells). However, this work is limited to performance tests under low mass loading, which cannot adequately assess the practical application prospects of the materials. Therefore, this study has not achieved a breakthrough in performance under practical conditions.

Response: Thank you for your constructive comment. We further evaluate the electrochemical performances of Fe||PTBA battery under practical conditions (e.g., tests under high mass loading and high areal capacity, validation in pouch cells), and added related discussion in the revised Manuscript: Fe||PTBA battery with $10.2 \text{ mg cm}^{-2}_{\text{PTBA}}$ still achieves high capacities of 229 mAh g^{-1} at 1 A g^{-1} and 147 mAh g^{-1} at 10 A g^{-1} (Supplementary Fig. S25), which is superior to recently reported values ($120/53 \text{ mAh g}^{-1}$ at $1/10 \text{ A g}^{-1}$ under $\sim 8 \text{ mg cm}^{-2}$)¹⁵. The desirable electrochemical performances highlight the electrochemical practical potential of phosphorus-engineered redox-active and stable organic materials towards better aqueous iron batteries. Besides, the assembled Fe||PTBA pouch-type cell even at a high mass loading of $10.7 \text{ mg cm}^{-2}_{\text{PTBA}}$ demonstrates a high areal capacity of 2.25 mAh cm^{-2} with 91.3% capacity retention after 1000 cycles at 1 A g^{-1} (Supplementary Fig. S26), demonstrating its great potential for practical applications.

Fig. S25. (a) Rate metrics and (b) cycling stability of Fe||PTBA battery with a high-mass loading of 10.2 mg cm⁻²_{PTBA}.

Fig. S26. Cycling stability of Fe||PTBA pouch cell at 1 A g⁻¹.

3. The authors should perform electrochemical performance tests under practical conditions (such as high mass loading, high areal capacity, and pouch cell configuration) to better evaluate the practical potential of the material.

Response: Thank you for your insightful comment. We supplement additional experiments to evaluate the electrochemical performances of Fe||PTBA battery under practical conditions (*e.g.*, tests under high mass loading and high areal capacity, validation in pouch cells), please see the detail shown above.

4. In Figure 1f, acetylene black conductive agent was added during the electronic conductivity test. However, electronic conductivity measurements should evaluate the intrinsic properties of the pure material without incorporating any additives.

Response: Thank you for your constructive comment. We have supplemented the electronic conductivities of pure PTBA and NTBA small molecules in Fig. S10, and added the discussion in the revised Manuscript: Compared with NTBA ($7.80 \times 10^{-9} \text{ S cm}^{-1}$), PTBA with the extended π -conjugated aromatic structure shows an enhanced electrical conductivity ($8.49 \times 10^{-9} \text{ S cm}^{-1}$, Supplementary Fig. S10). It is further boosted to 27.05 S cm^{-1} with the addition of 30 wt%

acetylene black conductive agent (Fig. 1f and Supplementary Fig. S11), facilitating efficient electron transfer.

Fig. S10. Electrical conductivities of PTBA and NTBA small molecules.

5. In Figure 3c, the original data should be included alongside the XPS spectra.

Response: Thank you for your valuable suggestion. We have included the original data alongside the XPS spectra in Fig. 3c in the revised Manuscript.

Fig. 3 (c) XPS spectra of C 1s corresponding to the selected voltages marked in the GCD curve.

6. In Figure S13, the schematic is inaccurate. Fe^{2+} intercalation can also occur in the PTBA cathode.

Response: Thank you for your insightful comment. We are sorry for this mistake. We have revised Fig. S13 to correctly display the schematic configuration of Fe||PTBA cell.

Fig. S13. Schematic configuration of Fe||PTBA cell, including Fe metal anode, aqueous 1 M Fe(OTF)₂ electrolyte, and PTBA cathode.

7. In Figure S14, it is reasonable that the PTBA cathode containing 30% acetylene black additive exhibits superior capacities compared to the 20% counterpart, which can be attributed to improved electronic conductivity. However, the fact that the PTBA cathode with 30% acetylene black additive delivers higher capacity than the 40% counterpart requires further explanation from the authors.

Response: Thank you for your insightful comment. We explained this point behind Fig. S14 in the revised Supplementary Information: Although increasing the content of conductive agent can improve the electrical conductivity of organic cathodes, it inevitably reduces the mass loading of active materials. This leads to a higher proportion of inactive components in the battery, thereby causing an unexpected reduction in capacity.^[S14]

Response to the Comments of Reviewer #2:

We would like to express our sincere thanks to the expert reviewer for the constructive and insightful comments which significantly contributed to improving the quality of the manuscript. Please find below a detailed response to each of the comments.

Reviewer #2: The manuscript entitled "Phosphorus-Activated Carboxyl Small Molecule Cathode for High-Capacity and Long-Life Iron-Organic Batteries" submitted by Zhang et al. presents the development of two organic molecules with bipolar features as cathode materials for aqueous iron-ion batteries. By modulating the p-type electron-donating phosphorus and tert-N motifs, 4,4',4''-phosphanetriyltribenzoic acid (PTBA) exhibits tailored charge distributions and energy levels, endowing the Fe||PTBA battery with state-of-the-art capacity (276 mAh g⁻¹) and outstanding

cycling stability (60,000 cycles) among reported cathode materials in IIBs. The authors investigated the Fe^{2+} ion storage pathways in the PTBA electrode through XPS, FTIR, and DFT calculations. However, the following questions need to be addressed.

1. What are the comparative advantages of iron-ion batteries over zinc-ion batteries in aqueous systems, based on the latest advancements in the field? These should be clarified in the Introduction section.

Response: Thank you for your insightful comment. We clarified this point in the Introduction section in the revised Manuscript: Among various aqueous batteries employing multivalent transition metal anodes (such as Zn, Mg, Al, Fe)⁷⁻¹¹, zinc-ion batteries are prevailing systems owing to their high theoretical capacity ($820 \text{ mAh g}^{-1}/5855 \text{ mAh cm}^{-3}$), low redox potential ($-0.76 \text{ V vs. standard hydrogen electrode}$) and relatively low cost ($\sim 2.5 \text{ USD per kg}$) of Zn anodes^{12,13}. In contrast, iron-ion batteries (IIBs) represent a compelling avenue for energy storage solutions, benefiting from the even higher theoretical capacity ($960 \text{ mAh g}^{-1}/7557 \text{ mAh cm}^{-3}$), moderate Fe^{2+}/Fe redox potential (-0.44 V vs. SHE) and ultra-low cost (about 40 times cheaper than Zn) of Fe anodes^{14,15}.

2. More evidence is required to demonstrate the contribution of carboxyl groups to the capacity of IIBs.

Response: Thank you for your constructive comment. Systematical characterizations including FT-IR spectra (Fig. 3b), XPS spectra (Fig. 3c-e and Fig. S27), electrochemical experiments (Fig. 2b and Fig. S18) and theoretical calculations (Fig. 5d-f) unravel the fact that PTBA cathode initiates a successive two-step 4 e^{-} redox reaction (Fig. 4g), which entails primarily 1 e^{-} phosphorus reaction with OTF^{-} ions, and then 3 e^{-} carboxyl redox with Fe^{2+} ions. This result suggests the dominant contribution of carboxyl groups to the capacity of IIBs.

We further studied this point with supplementary electrochemical experiments of TCB cathode (only containing carboxyl groups) in Fig. S18b, and added related discussion behind Fig. S18b in the revised Supplementary Information: TCB (containing carboxyl) and TPP cathodes (containing phosphorus) exhibit relatively low capacity of 121 mAh g^{-1} (66% carboxyl utilization) and 59 mAh g^{-1} (57.7% phosphorus utilization, Fig. S18), respectively, which can be further boosted to 276 mAh g^{-1} for PTBA cathode (containing carboxyl/phosphorus sites) with 98.5% utilization (Fig. 2b). This is because that PTBA cathode with P-extended conjugated structure

liberates the lowest energy gap (2.28 eV) compared with TCB/TPP cathode (2.71/3.16 eV, Fig. S9), which significantly increases the utilization of carboxyl redox sites. In addition, despite the $1 e^-$ capacity contribution of P site (vs. total $4 e^-$ charge storage), it boosts the average discharge voltage of PTBA to 0.8 V (Fig. 2b), significantly higher than TCB cathode (0.5 V). These results highlight the importance of P group in PTBA for promoting high voltage and high carboxyl utilization for enhanced capacity storage.

Fig. S18. (b) GCD curves of Fe||TCB battery in 1M Fe(OTF)₂/H₂O.

3. The GCD curves of Fe||TCB and Fe||TPP batteries should be provided.

Response: Thank you for your valuable suggestion. We supplemented the GCD curves of Fe||TCB and Fe||TPP batteries in Fig. S18, and improved the discussion in the revised Manuscript: Impressively, Fe||PTBA battery delivers superior cycling stability with 90.9% capacity retention after 5000 cycles at $0.5 A g^{-1}$ (Fig. 2g), far surpassing other Fe||NTBA, Fe||TCB and Fe||TPP cells (Supplementary Fig. S18 and Fig. S19).

Fig. S18. GCD curves of (a) Fe||TPP and (b) Fe||TCB batteries in 1M Fe(OTF)₂/H₂O.

4. The electrical conductivity of the electrode improves with increasing conductive carbon content, yet an unexpected reduction in capacity is observed, which contradicts prior findings. What underlying mechanisms could explain this phenomenon?

Response: Thank you for your insightful comment. We explained this point behind Fig. S14 in the revised Supplementary Information: Although increasing the content of conductive agent can improve the electrical conductivity of organic cathodes, it inevitably reduces the mass loading of active materials. This leads to a higher proportion of inactive components in the battery, thereby causing an unexpected reduction in capacity.^[S14]

5. The energy storage mechanism is investigated by the ex-situ XPS. The authors gave the variation of C1s XPS spectra during the charging/discharging process. However, the O 1s and P 2p spectra XPS spectra of the PTBA electrode are not given. Therefore, the O and P XPS spectra should be deconvoluted and analyzed to prove the reversible reaction in the PTBA structure during the charging/discharging process.

Response: Thank you for your valuable suggestion. We provided supplemental O 1s and P 2p XPS spectra of PTBA cathodes (Fig. S27) to prove the reversible reaction, and added the discussion in the revised Manuscript: Furthermore, XPS spectra of O 1s and P 2p species exhibit evolution trends consistent with C 1s spectra (Supplementary Fig. S27), collectively confirming the highly reversible redox activity of C=O and C–P groups in PTBA cathode.

Fig. S27. (a) O 1s and (b) P 2p XPS spectra of PTBA cathode corresponding to the selected voltages marked in the GCD curve.

6. Has the morphology of the PTBA material changed after long-term cycling?

Response: Thank you for your insightful comment. We provided high-resolution SEM images of PTBA cathodes before and after long-term cycling in Fig. S20, and added the discussion in the revised Manuscript: SEM images and XPS analysis reveal that PTBA cathode maintains nearly identical morphologies and redox-active functional groups after prolonged cycling (Supplementary Fig. S20 and Fig. S21), confirming its exceptional structural stability and electrochemical

reversibility.

Fig. S20. SEM images of PTBA cathodes in $\text{Fe}(\text{OTF})_2/\text{H}_2\text{O}$ electrolyte (a) before and (b) after long-term cycling.

7. It should be verified whether hydrogen ions participate in the energy storage process during the reaction.

Response: Thank you for your insightful comment. To confirm this point, we provided comparative electrochemical experiment in Fig. S28, and added the discussion behind Fig. S28 in the revised Supplementary Information: Comparing with 1 M $\text{Fe}(\text{OTF})_2/\text{H}_2\text{O}$ electrolyte (pH=3~4), $\text{Fe}||\text{PTBA}$ cell in $\text{HOTF}/\text{H}_2\text{O}$ electrolyte displays completely different electrochemical behaviors with a negligible capacity contribution of 14 mAh g^{-1} , indicating that H^+ ions hardly participate in the electrochemical reaction process. Meanwhile, the slight acidity and extremely low anion concentration of $\text{HOTF}/\text{H}_2\text{O}$ electrolyte fail to effectively activate the redox activity of p-type C-P groups, leading to insignificant capacity storage. Thus, the role of H^+ in the operation of $\text{Fe}||\text{PTBA}$ battery can be ignored.

Fig. S28. A GCD curve of $\text{Fe}||\text{PTBA}$ battery in $\text{HOTF}/\text{H}_2\text{O}$ electrolyte (with the same pH value of $\text{Fe}(\text{OTF})_2/\text{H}_2\text{O}$ electrolyte).

Response to the Comments of Reviewer #3:

We would like to express our sincere thanks to the expert reviewer for the constructive and insightful comments which significantly contributed to improving the quality of the manuscript. Please find below a detailed response to each of the comments.

Reviewer #3: The authors demonstrated phosphorus redox activity in a carboxyl small molecule, through designing two bipolar carboxyl organic small molecules, 4,4',4''-phosphanetriyltribenzoic acid (PTBA) and 4,4',4''-nitrilotribenzoic acid (NTBA) with tuned charge distributions and energy levels by incorporating the electron-donating p-type phosphorus and tert-N motifs as cathode materials for iron-organic batteries (IOBs). This study not only discovered new-type phosphorus redox chemistry in carboxyl cathode, but also solved the capacity and durability limitations of prevailing polyaniline cathodes, including state-of-the-art capacity (276 mAh g^{-1}) and outstanding cycling stability (60,000 cycles). This manuscript is of importance with well-organized analysis and discussion, and also highlights the potential of phosphorus-engineered redox-active and stable organic materials toward better IOBs. Thus, I recommend its publication in Nature Communications after a revision.

1. What are the theoretical capacities of PTBA and NTBA? Is the ultrahigh utilization rate of 98.5% the actual utilization efficiency?

Response: Thank you for your insightful comment. We provided the theoretical capacities of PTBA and NTBA in the revised Manuscript, Methods, Electrochemical Tests: When considering the molecular weight of 394.3 g mol^{-1} and four-electron transfer during the redox process, the theoretical capacity is calculated to be 271.9 mAh g^{-1} for PTBA cathode. Similarly, the theoretical capacity of NTBA is 284.1 mAh g^{-1} . Given the negligible capacity contributions from acetylene black (8 mAh g^{-1} , Supplementary Fig. S16), the real capacity of PTBA cathode of 268 mAh g^{-1} is calculated to be approximately $4e^-$ redox reactions, corresponding to an actual utilization efficiency of 98.5% for n-p fused carboxyl/phosphorus motifs.

2. Why choose $\text{Fe}(\text{OTf})_2/\text{H}_2\text{O}$ solution as the electrolyte? The reason behind this behavior should be explained. Is PTBA cathode also applicable to the cheaper $\text{FeSO}_4/\text{H}_2\text{O}$ electrolyte?

Response: Thank you for your constructive comment. We clarified this point with supplemental experiments (Fig. S15) of PTBA cathode in $1 \text{ M FeSO}_4/\text{H}_2\text{O}$ electrolyte, and added the discussion behind Fig. S15 in the revised Supplementary Information: For a comparison, we studied the

electrochemical properties of Fe||PTBA battery in 1 M FeSO₄/H₂O electrolyte (Fig. S15). Compared with Fe(OTF)₂/H₂O electrolyte, FeSO₄/H₂O electrolyte shows relatively low surface wettability (Fig. S15a) and ionic conductivity (Fig. S15b). As well-established, CF₃SO₃⁻ ions demonstrate weaker solvation interactions with H₂O and metal ions than SO₄²⁻ anions, which promote efficient desolvation to liberate enhanced reaction kinetics and electrochemical activity (*Nat. Commun.* **2023**, *14*, 3117).^[S16] As a result, PTBA cathode in FeSO₄/H₂O electrolyte delivers similar but slightly lower capacity, rate performance, and cycling stability (Fig. S15c–e) than that of Fe(OTF)₂/H₂O electrolyte. Such a result suggests the better adaptability of PTBA cathode in Fe(OTF)₂/H₂O electrolyte, which thus is selected as the electrolyte.

Fig. S15. (a) Contact angles and (b) ion conductivities of two different electrolytes. (c) A GCD curve, (d) rate capacities, (e) cycling stability of Fe||PTBA battery in 1 M FeSO₄/H₂O electrolyte.

3. Given the inherent irreversibility of iron plating/stripping processes, which are invariably accompanied by parasitic reactions. So how can its outstanding cycle stability (60,000 cycles, Fig. 2h) be explained?

Response: Thank you for your insightful comment. We explained this point behind Fig. S20 in the revised Supplementary Information: Of note, during the cycling process of Fe||PTBA battery, a thick glass fiber separator (50 μm) was used to alleviate the problem of inherent irreversibility and parasitic reactions in iron foil anode (with a thickness of 50 μm) to achieve outstanding cycle stability after 60,000 cycles. Such favorable electrochemical processes allow the high compatibility between PTBA cathode and Fe anode for propelling superior Fe-organic batteries. Besides, some recent studies about aqueous Fe-ion batteries have demonstrated excellent cycling stability, such

as *Nat. Commun.* **2023**, *14*, 3117 (39,000 cycles); *Energy Environ. Sci.*, **2025**, *18*, 1428–1439 (27,000 cycles), highlighting the efficient plating/stripping processes of Fe anodes.

4. Stable electrochemical performance in high-mass-loading organic electrodes ($>10 \text{ mg cm}^{-2}$) is critical for practical battery applications. High-quality cycling data for PTBA cathodes under these conditions would be particularly compelling, as it could pave the way for iron-organic batteries with improved large-current durability.

Response: Thank you for your valuable suggestion. We studied the electrochemical performance of high-mass-loading PTBA cathode in Fig. S25, and added the discussion in the revised Manuscript: Fe||PTBA battery with 10.2 mg cm^{-2} PTBA still achieves high capacities of 229 mAh g^{-1} at 1 A g^{-1} and 147 mAh g^{-1} at 10 A g^{-1} (Supplementary Fig. S25), which is superior to recently reported values ($120/53 \text{ mAh g}^{-1}$ at $1/10 \text{ A g}^{-1}$ under $\sim 8 \text{ mg cm}^{-2}$)¹⁵. The desirable electrochemical performances highlight the electrochemical practical potential of phosphorus-engineered redox-active and stable organic materials towards better aqueous iron batteries.

Fig. S25. (a) Rate metrics and (b) cycling stability of Fe||PTBA battery with a high-mass loading of 10.2 mg cm^{-2} PTBA.

5. Given the claimed outstanding stability of PTBA cathode, so it is recommended to supplement the comparative XPS analysis before and after prolonged cycling.

Response: Thank you for your valuable suggestion. We supplemented the data in Fig. S21, and added the discussion in the revised Manuscript: SEM images and XPS analysis reveal that PTBA cathode maintains nearly identical morphologies and redox-active functional groups after prolonged cycling (Supplementary Fig. S20 and Fig. S21), confirming its exceptional structural stability and electrochemical reversibility.

Fig. S21. C 1s XPS spectra of PTBA cathodes before and after prolonged cycling.

6. Essential battery fabrication parameters require clarification, particularly regarding Fe anode diameter, separator specifications and electrolyte content. Additionally, the initial open-circuit voltage measurement for a newly-assembled coin cell should be reported.

Response: Thank you for your insightful comment. We have provided experimental details in the revised Manuscript, Methods, Electrochemical Tests: To construct aqueous iron-organic batteries (IOBs), PTBA (or NTBA, TCB, TPP) cathode, 1 mol L⁻¹ aqueous Fe(OTF)₂ electrolyte (135 μL), Fe metal anode (> 99.99% purity, 1.6 cm in diameter), and glass fiber separator (Whatman) were packaged into 2032 coin-type cells. Additionally, the initial open-circuit voltage is ~0.7 V for a newly-assembled Fe||PTBA coin cell.

According to the comments of the expert reviewers, we have thoroughly revised and improved our manuscript to meet the requirements of the journal. Now we submit the revised version for your kind consideration.

We would like to express our sincere thanks to you and the expert reviewers for the constructive and positive comments which significantly contributed to improving the quality of the manuscript.

Yours sincerely,

Prof. Dr. Mingxian Liu

School of Chemical Science and Engineering

Tongji University